# An evolutionary mechanism to assimilate new nutrient sensors into the mTORC1 pathway

Grace Y. Liu [1,2,3] ✉, Patrick Jouandin [4,5,7], Raymond E. Bahng[1,2,3], Norbert Perrimon [4,5] ✉ & David M. Sabatini [6] ✉

Animals sense and respond to nutrient availability in their environments, a task coordinated in part by the mTOR complex 1 (mTORC1) pathway. mTORC1 regulates growth in response to nutrients and, in mammals, senses specific amino acids through specialized sensors that bind the GATOR1/2 signaling hub. Given that animals can occupy diverse niches, we hypothesized that the pathway might evolve distinct sensors in different metazoan phyla. Whether such customization occurs, and how the mTORC1 pathway might capture new inputs, is unknown. Here, we identify the *Drosophila melanogaster* protein Unmet expectations (CG11596) as a species-restricted methionine sensor that directly binds the fly GATOR2 complex in a fashion antagonized by S-adenosylmethionine (SAM). We find that in Dipterans GATOR2 rapidly evolved the capacity to bind Unmet and to thereby repurpose a previously independent methyltransferase as a SAM sensor. Thus, the modular architecture of the mTORC1 pathway allows it to co-opt pre-existing enzymes to expand its nutrient sensing capabilities, revealing a mechanism for conferring evolvability on an otherwise conserved system.

By detecting features of their environments (signals), the sensory systems of eukaryotes confer advantages for survival and reproduction. These signals are often specialized and reflect the specific biochemical and biophysical properties of the niche of each organism. To detect new signals over the course of evolution, sensory systems must acquire novel receptors and link these to the pre-existing pathways that actuate behavioral or metabolic changes. With few exceptions, how conserved signaling networks evolve mechanisms to detect new inputs is poorly understood[1–3].

In some sensory systems, this capacity arises through the duplication of existing receptors, followed by the modification of the newly formed paralogs to increase promiscuity or alter substrate preferences (Fig. 1a). For example, successive expansion and mutation of certain receptor classes—including some hormone receptors, olfactory receptors, Toll-like receptors, and TRP ion channels—has expanded the complexity of chemosensation in different species[2,4–7]. However, although this strategy expands the ligand or activity space for receptors that are already connected to a pathway, it is a poor model for

[1]Whitehead Institute for Biomedical Research and Massachusetts Institute of Technology, Department of Biology, 455 Main Street, Cambridge, MA, USA. [2]Department of Biology, Massachusetts Institute of Technology, 77 Massachusetts Avenue, Cambridge, MA, USA. [3]Koch Institute for Integrative Cancer Research and Massachusetts Institute of Technology, Department of Biology, 77 Massachusetts Avenue, Cambridge, MA, USA. [4]Department of Genetics, Blavatnik Institute, Harvard Medical School, Boston, MA, USA. [5]Howard Hughes Medical Institute, Harvard Medical School, Boston, MA, USA. [6]Institute of Organic Chemistry and Biochemistry, Flemingovo n. 2, 166 10 Praha 6, Prague, Czech Republic. [7]Present address: Institut de Recherche en Cancérologie de Montpellier, Inserm U1194-UM-ICM, Campus Val d'Aurelle, Montpellier, Cedex 5, France. ✉e-mail: grace.yun.liu@berkeley.edu; perrimon@genetics.med.harvard.edu; david.sabatini@uochb.cas.cz

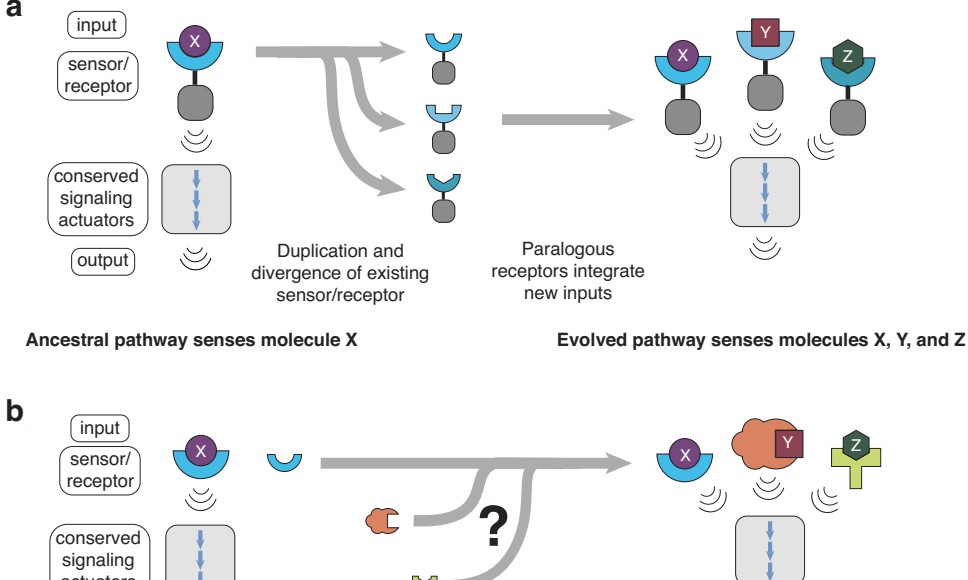

**Fig. 1 | mTORC1 nutrient sensing is a model system for interrogating how conserved signaling pathways evolve new sensory inputs through novel molecular partnerships. a** Classical sensory systems evolve new functional inputs by altering the ligand-binding capabilities of an existing sensor or receptor, often after duplication of the receptor. This evolutionary strategy gives rise to families of paralogous receptors that signal to conserved downstream actuators through shared domains. **b** Some non-canonical sensory systems, such as the mTORC1 pathway, use sets of unrelated proteins as sensors/receptors. These receptors may have evolved from nonsensory precursors, and it is not known how they forged new molecular interactions with conserved components of the pathway.

receptors that emerge through novel molecular partnerships (Fig. 1b). A key question, therefore, is how functional diversification occurs in the absence of paralogous duplication. What evolutionary strategies are employed by signaling networks that evolve multiple *unrelated* receptors to sense new inputs?

The mechanistic target of rapamycin complex 1 (mTORC1) pathway is a model for this latter type of network. The mTORC1 pathway surveys the concentrations of nutrients, such as amino acids and related metabolites, to regulate growth and metabolism[8–12]. Upon activation by nutrients, mTORC1 allocates cellular resources towards anabolism by promoting protein and lipid biosynthesis and inhibiting autophagy. Because organisms have a wide range of lifestyles and diets, we postulate that the mTORC1 pathway is under pressure to evolve receptors for the most important nutrients within a given niche. In mammals, these receptors take the form of specialized nutrient sensors—Sestrin2, CASTOR1, SAMTOR, and LYCHOS—that bind, respectively, to leucine, arginine, the methionine-derived methyl donor *S*-adenosylmethionine (SAM), and cholesterol[13–16]. When cells are starved of nutrients, the mammalian nutrient sensors interact with several conserved protein complexes that relay signals to control mTORC1 kinase activity. These complexes, which comprise the core nutrient sensing machinery of the mTORC1 pathway, regulate the Rag GTPases and include the large GATOR1 and GATOR2 complexes, as well as KICSTOR, a vertebrate-specific partner of GATOR1[17–19]. Upon nutrient replenishment, the sensors bind their cognate metabolites, which releases the sensors from the core complexes and thus reactivates mTORC1[20,21].

Although the general architecture of the mTORC1 pathway is conserved across eukaryotes, the pathway must remain sufficiently flexible to accommodate organisms with distinct nutritional needs. Unlike most of the core components of the mTORC1 pathway, which are present from yeast to humans, the mammalian nutrient sensors are only sporadically conserved in metazoans and completely absent from yeast[22,23]. Genomic analyses reveal that *D. melanogaster* lacks

homologs of the mammalian arginine sensors but retains genes for both a full Sestrin protein and a substantially truncated SAMTOR protein; by contrast, *C. elegans* possesses homologs of Sestrin and the lysosomal arginine sensor SLC38A9 while lacking a clear SAMTOR equivalent[24]. Despite their similar modes of action, the known mammalian nutrient sensors bear no homology to each other. Based on these observations, we propose that nutrient sensors comprise a plastic regulatory layer atop the conserved core of the mTORC1 pathway machinery—one that can be customized to detect limiting nutrients in different metazoan phyla[8,25].

To understand whether and how the mTORC1 pathway acquires custom nutrient sensors, we searched for novel sensors in *Drosophila melanogaster*, an organism that shares many pathway components with humans but consumes a divergent diet. We discover a new species-restricted SAM sensor and follow its evolutionary history to pry open the structural logic of the nutrient-sensing axis. We show that this sensor, which we named Unmet expectations, is an evolutionary intermediate, caught between its ancestral enzymatic function and a recently acquired role in the mTORC1 pathway. By comparing SAM sensing in different clades, we find that flies and vertebrates independently evolved unrelated, mechanistically distinct sensors that converge upon the same metabolite. Unexpectedly, our results shed light on the origins of the nutrient sensors and reveal remarkable features of GATOR2, a core signaling hub for the mTORC1 pathway, that allow the pathway to rapidly co-opt ligand-binding proteins and adapt to metabolic niches across evolution.

## Results

### Unmet binds to fly GATOR2 in an *S*-adenosylmethionine (SAM)-regulated fashion

The GATOR complexes have emerged as central integrators of metabolic information for the mTORC1 pathway. To identify novel nutrient sensors, we searched for GATOR-binding partners in *Drosophila melanogaster*. We generated anti-FLAG immunoprecipitates from *D.*

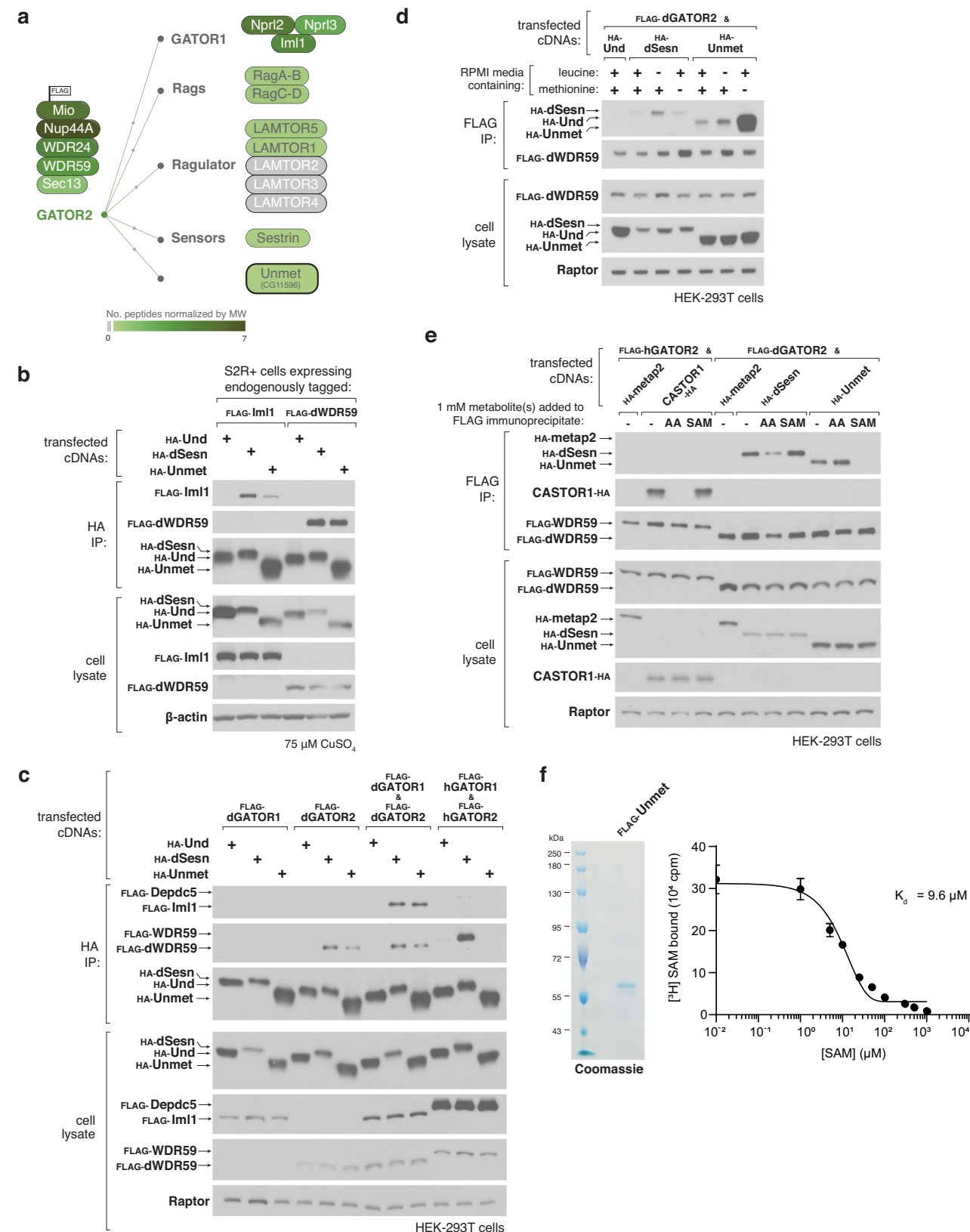

melanogaster Schneider 2 (S2R +) cells expressing FLAG-tagged Mio, a core component of the *Drosophila* GATOR2 (dGATOR2) complex. Mass spectrometry analyses revealed that beyond capturing other components of the dGATOR complexes and the leucine sensor dSestrin, these immunoprecipitates also contained the previously uncharacterized fly protein CG11596, which we have renamed Unmet

expectations (Unmet) for reasons described below (Fig. 2a). When transiently expressed in S2R+ cells, HA-tagged Unmet robustly co-immunoprecipitated endogenous dGATOR2, as detected via its dWDR59 component, as well as the dGATOR1 complex, as detected via its Iml1 component (Fig. 2b). Because the dGATOR1 and dGATOR2 complexes appear to be more tightly associated in flies than in

**Fig. 2 | SAM regulates the interaction between the *D. melanogaster* protein Unmet expectations and the fly GATOR2 complex. a** Mass spectrometric analyses identify Unmet-derived peptides in immunoprecipitates from S2R+ cells expressing FLAG-tagged Mio, a component of the dGATOR2 complex. Unmet and previously known components of the mTORC1 pathway are colored by normalized peptide representation according to the scale below. **b** Recombinant Unmet co-immunoprecipitates endogenous GATOR1 and GATOR2 components in S2R+ cells. Anti-HA immunoprecipitates were prepared from S2R+ cells bearing endogenous FLAG knock-in tags at either the Iml1 (dGATOR1) or the dWDR59 (dGATOR2) locus, and transfected with the indicated cDNAs in copper-inducible metallothionein (MT) expression vectors. Following a 48-h induction with 75 μM CuSO₄, cell lysates and immunoprecipitates were analyzed by immunoblotting for levels of the relevant epitope tags. HA-Und served as a negative control. **c** Recombinant Unmet interacts with dGATOR2, but not dGATOR1 or the corresponding human complexes. Anti-HA immunoprecipitates were collected from HEK-293T cells co-transfected with the indicated cDNAs in expression vectors and analyzed alongside cell lysates as in (**b**). **d** Deprivation of methionine, but not leucine, enhances the interaction between Unmet and dGATOR2. HEK-293T cells transiently expressing FLAG-tagged dGATOR2 and the indicated HA-tagged cDNAs were cultured in full RPMI or RPMI lacking leucine or methionine for 1 h. FLAG immunoprecipitates and cell lysates were analyzed by immunoblotting for the levels of the relevant proteins. **e** SAM, but not amino acids, disrupts the interaction between Unmet and dGATOR2 in vitro. FLAG immunoprecipitates were prepared from HEK-293T cells transfected with the indicated cDNAs. A mixture containing 1 mM of each amino acid or 1 mM of SAM was added directly to the immunoprecipitates. FLAG immunoprecipitates and cell lysates were analyzed as in (**d**). **f** Unmet binds SAM with a $K_d$ of 9.6 μM. Purified FLAG-Unmet protein was analyzed by SDS-polyacrylamide gel electrophoresis followed by Coomassie blue staining. Binding assays contained 10 μg of purified FLAG-Unmet, 5 μM [³H]SAM, and the indicated concentrations of unlabeled SAM. Values for each point represent the means ± s.d. of three technical replicates from one representative experiment. Binding experiments were repeated three times.

mammalian systems, these data are consistent with Unmet binding to either or both of the dGATOR complexes. To differentiate between those possibilities, we transiently co-expressed the dGATOR1 and/or dGATOR2 complexes with Unmet in human embryonic kidney 293 T (HEK-293T) cells (Fig. 2c). Like dSestrin, which has been characterized as a GATOR2-binding protein, Unmet co-immunoprecipitated dGATOR2, but not dGATOR1, in this reconstituted system. Unmet, therefore, binds to dGATOR2 without requiring any additional *Drosophila*-specific factors.

The Unmet protein sequence possesses an N2227 domain, which defines homologs from yeast to human and may contain methyltransferase activity[26]. Indeed, recent work has shown that the human, rat, chicken, and *Saccharomyces cerevisiae* orthologs of Unmet are all capable of methylating the histidyl ring of the dipeptide L-carnosine to produce anserine, albeit at low catalytic efficiencies[26,27]. Despite strong sequence conservation at the putative small molecule binding sites (Supplementary Fig. 1a), it is unknown whether Unmet retains this activity. Moreover, it is unclear whether such activity, even if present, would be functionally relevant in flies, as carnosine and anserine are reported to be nearly absent from *Drosophila* tissues[28].

Given the conservation of Unmet between flies and humans, we tested the capacity of Unmet to bind to the human GATOR2 complex. Unlike fly Sestrin, Unmet did not interact with transiently expressed or endogenous human GATOR2 (Fig. 2c, Supplementary Fig. 2a). These results indicate that the interaction between Unmet and dGATOR2 is not conserved in vertebrates and may instead be specific to the fly lineage.

Previous studies have shown that homologs of Unmet directly bind to the methionine-derived methyl donor SAM through their N2227 domains[27]. By analogy to the amino acid sensors Sestrin and CASTOR1, which contain small molecule binding sites and dissociate from GATOR2 in the presence of specific amino acids, we postulated that small molecules might also modulate the Unmet-dGATOR2 interaction. Consistent with this hypothesis, withdrawal of the amino acid methionine, but not leucine, from the culture medium enhanced the interaction of recombinant Unmet with dGATOR2 in both HEK-293T and S2R+ cells (Fig. 2d, Supplementary Fig. 2b).

To determine whether methionine acts directly on Unmet—as leucine and arginine do on Sestrin2 and CASTOR1, respectively—or whether the interaction is mediated by a related metabolite, as with SAMTOR, we immunopurified the Unmet-dGATOR2 complex from amino acid-starved cells. Addition of a cocktail of amino acids to lysates disrupted the CASTOR1-human GATOR2 and dSestrin-dGATOR2 complexes but did not release Unmet from dGATOR2. Instead, SAM, which had no effect on the CASTOR1 and dSestrin interactions with GATOR2, robustly dissociated Unmet from dGATOR2 (Fig. 2e).

Because the human homolog of Unmet has been co-crystallized with carnosine and various derivatives of SAM[27], we tested whether these small molecules could perturb the interaction between Unmet and dGATOR2. Unlike SAM, which dissociated the Unmet-dGATOR2 complex in a dose-dependent manner, carnosine or S-adenosylhomocysteine (SAH), the demethylated form of SAM, had no effect (Supplementary Fig. 2c). Despite the discrepancy between the impact of SAM and SAH, SAM-dependent dissociation of Unmet from dGATOR2 is unlikely to require a methylation event, as the SAH analog sinefungin (SFG) is capable of breaking the Unmet-dGATOR2 interaction (Supplementary Fig. 2c).

Using an equilibrium binding assay similar to those previously used for analyses of Sestrin2 and SAMTOR, we found that radiolabeled SAM binds directly to purified Unmet. Excess cold SAM fully competed off the tritiated SAM, yielding a dissociation constant of 9.6 μM (Fig. 2f). Although SAH does not disrupt the interaction between Unmet and dGATOR2, it readily competes with labeled SAM for binding to Unmet (Supplementary Fig. 2d). These results suggest that Unmet binds both SAM and SAH but undergoes a conformational change to evict dGATOR2 only when a methyl-like moiety occupies the metabolite-binding cleft. In line with this hypothesis, sinefungin, which replaces the sulfonium (S-CH₃) group of SAM with a primary amine, also competes with labeled SAM for binding to Unmet and, as described above, displaces dGATOR2, while carnosine does not (Supplementary Fig. 2d). How the Unmet-dGATOR2 complex discriminates between SAM/SFG and SAH remains an open question, as all three metabolites likely bind to the same site on Unmet.

**Unmet confers methionine sensitivity on the fly TORC1 pathway**
Given that SAM binds Unmet and regulates its interaction with dGATOR2, we reasoned that Unmet might affect the ability of the *Drosophila* TORC1 (dTORC1) pathway to respond to methionine deprivation. Indeed, depletion of *unmet* mRNA by double-stranded RNA (dsRNA)-mediated RNA interference rendered the dTORC1 pathway insensitive to methionine starvation in S2R+ cells (Fig. 3a). Although dTORC1 responds to a different set of environmental amino acids than mammalian mTORC1, the effects of the *unmet* knockdown were remarkably specific. As detected by the phosphorylation of dS6K at residue Thr³⁹⁸, the dsRNA targeting *unmet* prevented dTORC1 inhibition upon methionine starvation while leaving leucine (Supplementary Fig. 3a), threonine, glutamine, phenylalanine, and tryptophan sensitivity intact (Fig. 3a).

We confirmed and extended this result using an orthogonal method for controlling *unmet* expression. To tune Unmet protein levels, we engineered an S2R+ cell line with a copper-inducible metallothionein (MT) promoter and a FLAG epitope tag knocked into the endogenous *unmet* locus, such that FLAG-Unmet expression responded to the concentration of copper sulfate in the culture

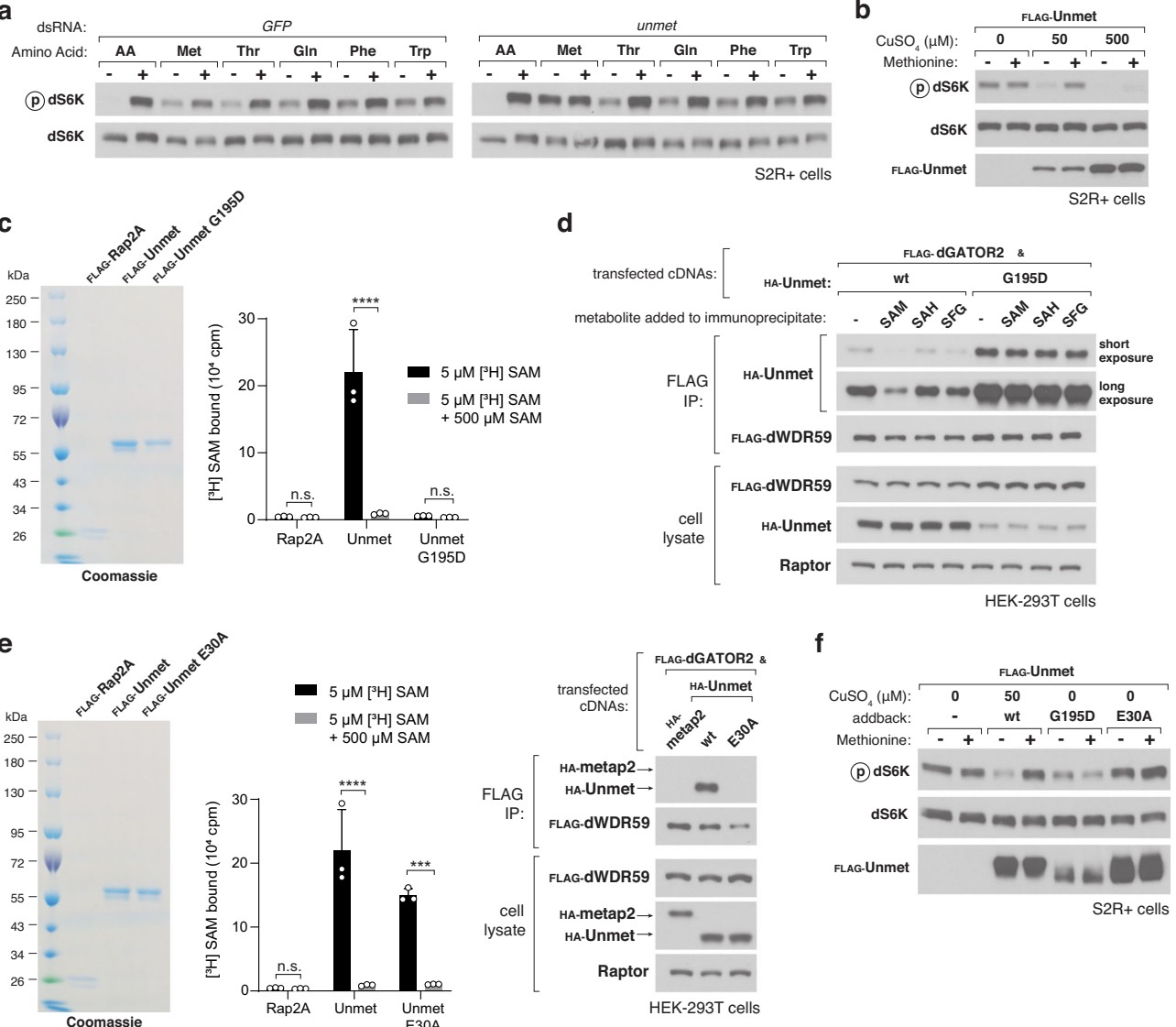

**Fig. 3 | Unmet signals methionine sufficiency to dTORC1 by acting as a negative regulator of the pathway. a** S2R+ cells were transfected with dsRNAs targeting either a control mRNA (*GFP*) or *unmet* mRNA. dsRNA-treated cells were then starved of the indicated amino acids for 90 min or starved and restimulated for 15 min. Cell lysates were analyzed by immunoblotting for the phosphorylation states and levels of the indicated proteins. **b** S2R+ cells expressing a copper-inducible FLAG-tagged Unmet from the endogenous locus were incubated with the indicated concentrations of CuSO₄ for 72 h. Cells were then starved of methionine or starved and restimulated, and cell lysates analyzed as in (**a**). **c** The G195D mutant of Unmet does not bind SAM. Binding assays were performed and immunoprecipitates analyzed as in Fig. 2f. Two-way ANOVA followed by Tukey's multiple comparison test; from left to right: adjusted $P = 1.0$; ****$P < 0.0001$; $P = 1.0$; n.s., not significant. Error bars represent the s.d. around the mean of three independent samples. Binding data for Rap2A and wild-type Unmet are shown again for clarity in (**e**). **d** The interaction between Unmet G195D and dGATOR2 is insensitive to SAM

and SFG. FLAG immunoprecipitates were prepared from HEK-293T cells transfected with the indicated cDNAs. 1 mM of the indicated metabolite was added directly to the immunoprecipitates. FLAG immunoprecipitates and cell lysates were analyzed as in Fig. 2d. **e** The E30A mutant of Unmet does not interact with dGATOR2 but maintains its SAM-binding capacity. Binding assays were performed and analyzed as in (**c**). Two-way ANOVA followed by Tukey's multiple comparison test; adjusted ***$P = 3.0 \times 10^{-4}$. Error bars represent the s.d. around the mean of three independent samples. For the Western blot, FLAG immunoprecipitates were prepared and analyzed as in (**d**). **f** S2R+ cells expressing a copper-inducible FLAG-tagged Unmet from the endogenous locus were engineered to stably express the indicated FLAG-tagged proteins. Cells were then induced for 72 h with either no CuSO₄, to mimic an *unmet*-null cell, or 50 μM CuSO₄, to mimic wild-type expression of Unmet. Cells were starved of methionine or starved and restimulated, and lysates were analyzed as in (**b**).

medium. In the absence of copper induction, *unmet* mRNA levels dropped more than 10-fold from the endogenous ones, mimicking an *unmet* knockdown (Supplementary Fig. 3b). Under those conditions, the dTORC1 pathway was wholly resistant to methionine starvation. Low induction (at 50 μM CuSO₄) of FLAG-Unmet restored the methionine responsiveness of the dTORC1 pathway, while substantial overexpression (at 500 μM CuSO₄) blunted dTORC1 activity (Fig. 3b). These data show that Unmet inhibits dTORC1 signaling in the absence

of methionine and, like CASTOR1 and Sestrin2 in human cells, suppresses the pathway when overexpressed.

Our finding that Unmet conveys methionine availability to the dTORC1 pathway led us to reevaluate the role of another putative fly SAM sensor[29]. Although the fly homolog of SAMTOR (dSAMTOR) is about 12 kDa smaller and only loosely conserved from its mammalian counterpart, we previously reported that dsRNA-mediated knockdown of *dSAMTOR* in fly cells abrogated dTORC1 inhibition upon withdrawal

of methionine[15]. To our surprise, however, attempts to reproduce this observation with the original dsRNA yielded inconsistent results, while a different dsRNA robustly lowered *dSAMTOR* mRNA levels without affecting methionine signaling (Supplementary Fig. 3c). To circumvent dsRNA-mediated artifacts, we introduced a copper-inducible promoter at the endogenous *dSAMTOR* locus in S2R+ cells. We then deprived the cells of copper to generate a *dSAMTOR*-null state. These cells showed no detectable *dSAMTOR* expression but remained sensitive to methionine (Supplementary Fig. 3c). Importantly, the methionine sensitivity of the dTORC1 pathway in *dSAMTOR*-null cells could be abolished by dsRNA-mediated knockdown of *unmet*. Overexpression of *dSAMTOR* failed to suppress dTORC1 activity or alter the methionine sensitivity of the pathway (Supplementary Fig. 3d), and consistent with the absence in flies of the KICSTOR complex—an obligate binding partner of human SAMTOR—dSAMTOR did not interact with either endogenous dGATOR1 or dGATOR2 (Supplementary Fig. 3e). Finally, *dSAMTOR*[−/−] larvae fed a methionine-free diet retain the capacity to inhibit dTORC1 (Supplementary Fig. 3f). While these data do not preclude dSAMTOR from acting on dTORC1 through other mechanisms, they suggest that our initial proposal as a component of the dTORC1 pathway in flies may have been due to misleading off-target effects of the particular dsRNA used at the time. We therefore conclude that Unmet, rather than dSAMTOR, is the relevant mediator of methionine sensing for the dTORC1 pathway. The function of dSAMTOR in fly cells, however, remains unknown.

If Unmet is required for dTORC1 to sense methionine, does its SAM-regulated interaction with dGATOR2 transduce that signal? To decouple the metabolite-binding capacity of Unmet from its ability to bind dGATOR2, we performed structure-guided mutagenesis of the protein. A glycine-to-aspartate replacement at the highly conserved G195 residue in the SAM-binding pocket of Unmet abolished its ability to bind SAM in vitro (Fig. 3c). The G195D SAM-binding mutant interacted robustly with dGATOR2 in a constitutive fashion (Fig. 3d). Using alanine scanning mutagenesis of surface-exposed residues, as inferred from the crystal structure of the human homolog of Unmet, we also identified a mutation at residue E30 of Unmet that disrupted its interaction with dGATOR2 without impairing its SAM-binding capacity (Fig. 3e).

To assess the effect of these Unmet mutants on dTORC1 signaling, we expressed the SAM-binding (G195D) and dGATOR2-binding (E30A) mutants in the S2R+ cells with copper-inducible expression of FLAG-Unmet (Fig. 3f). In the absence of copper, which leads to an Unmet-null state, the dTORC1 pathway in these cells is insensitive to methionine deprivation. Although expression of wild-type Unmet restored the methionine sensitivity of the pathway, expression of the G195D mutant constitutively inhibited dTORC1 signaling, suggesting that SAM must be able to bind to Unmet in order to activate the pathway. Meanwhile, expression of Unmet E30A had no effect on dTORC1 activity, demonstrating that the interaction between Unmet and dGATOR2 is required for dTORC1 to sense the absence of methionine and SAM. Thus, we conclude that Unmet conveys methionine levels to dTORC1 in cells in culture.

## Loss of Unmet in flies impairs organismal adaptation to methionine-restricted diets

To determine whether Unmet serves a corresponding function in vivo, we generated an *unmet*[−/−] mutant fly strain using CRISPR-Cas9-mediated deletion of the gene locus. *unmet*[−/−] flies had no detectable *unmet* mRNA (Supplementary Fig. 4a) but remained fully viable. However, unlike wild-type larvae, which showed blunted dTORC1 activity in the fat body after 24 h on a methionine-free diet, *unmet*[−/−] larvae failed to inhibit the dTORC1 pathway upon methionine starvation (Fig. 4a). This phenotype in the larval fat body, a homogeneous tissue amenable to biochemical analysis, recapitulates the signaling defect seen in cultured *unmet* knockdown cells. Loss

of *unmet* had no effect on dTORC1 signaling in larvae fed a full diet, indicating that the link between Unmet and dTORC1 is nutrient-dependent.

Guided by the expression pattern of *unmet*, which showed it to be highly enriched in the ovary of the adult female (Fig. 4b, Supplementary Fig. 4b), we sought to define a physiological requirement for SAM-sensing by the dTORC1 pathway. The *Drosophila* ovary is a nutrient-responsive tissue comprised of ovarioles, strings of egg chambers that proceed from a germarium through progressively more mature stages of development. Because egg production is so energy- and resource-intensive, oogenesis halts under protein starvation or prolonged stress[30–32]. To avoid investments in eggs or progeny that will not be viable, vitellogenic (yolk-forming) mid-stage egg chambers (stages 8–10) and some germline cysts undergo apoptosis in the ovaries of starved flies[31]. However, likely to ensure rapid reestablishment of egg production after permissive conditions are restored, early (stage 1–7) egg chambers are protected from apoptosis during starvation, slowing their growth but remaining intact and so preserving future female fertility[31,33,34].

Survival of early egg chambers in starved flies requires finely-tuned control of the dTORC1 pathway. In flies fed an amino-acid-free diet, ovarian-specific knockdown of negative regulators of dTORC1, including the dGATOR1 components dNprl2 and dNprl3, produces a sharp increase in apoptotic early egg chambers, suggesting that failure to downregulate dTORC1 signaling during early oogenesis triggers cell death under amino acid limitation[33]. Interestingly, single-cell sequencing of the fly ovary shows that expression of *unmet* is concentrated in young germ cells within the germarium (Fig. 4c), overlapping strongly with the cell populations that express *dNprl2* and *dNprl3* (Supplementary Fig. 4c)[35]. The fly ovary is also well-validated as a methionine-sensitive niche, with lifetime egg production tied to methionine availability[36,37]. Indeed, methionine supplementation alone is sufficient to restore fecundity in flies during dietary restriction, indicating that methionine may be a limiting nutrient for ovarian function[36]. We therefore hypothesized that Unmet contributes to the maintenance of early egg chambers under methionine and SAM restriction.

To test this model, we placed control or *unmet*[−/−] flies on either a full diet or a chemically-defined diet lacking methionine. After 1 or 5 days on this diet, ovaries were dissected and stained for the apoptosis factor cleaved *Drosophila* caspase 1 (Dcp-1) (Fig. 4d). Methionine starvation increased the number of degenerating early egg chambers in *unmet*[−/−] flies but not in the background-matched control, with the longer starvations enhancing the severity of the phenotype (Fig. 4e, f, Supplementary Fig. 4d). By contrast, methionine-starved mid-stage egg chambers underwent apoptosis at identical rates between *unmet*[−/−] and control flies (Supplementary Fig. 4e). Rapamycin treatment substantially rescued early egg chamber viability in methionine-starved *unmet*[−/−] flies, indicating that Unmet exerts a protective function under these conditions by suppressing dTORC1 signaling (Fig. 4g, Supplementary Fig. 4f). Following fly community convention, we have renamed *CG11596* as *unmet expectations*, because loss-of-function flies fail to sense and anticipate low-methionine (un-Met) conditions, leading to degradation of the female germline.

Taken in sum, these data converge upon a model in which Unmet detects drops in SAM levels within the germ cell environment and downregulates dTORC1 to prevent damage to early egg chambers (Fig. 4h). Loss of Unmet permits aberrant activation of dTORC1 under methionine restriction, triggering apoptosis in early egg chambers and compromising germline integrity (Fig. 4h). Degradation of early egg-chambers, in turn, impairs egg production up to weeks after restoration of a rich diet and can permanently reduce fertility[33]. The evolutionary acquisition of a SAM sensor may have conferred selective advantages by allowing flies to use a critical nutrient to gate reproductive investment[38].

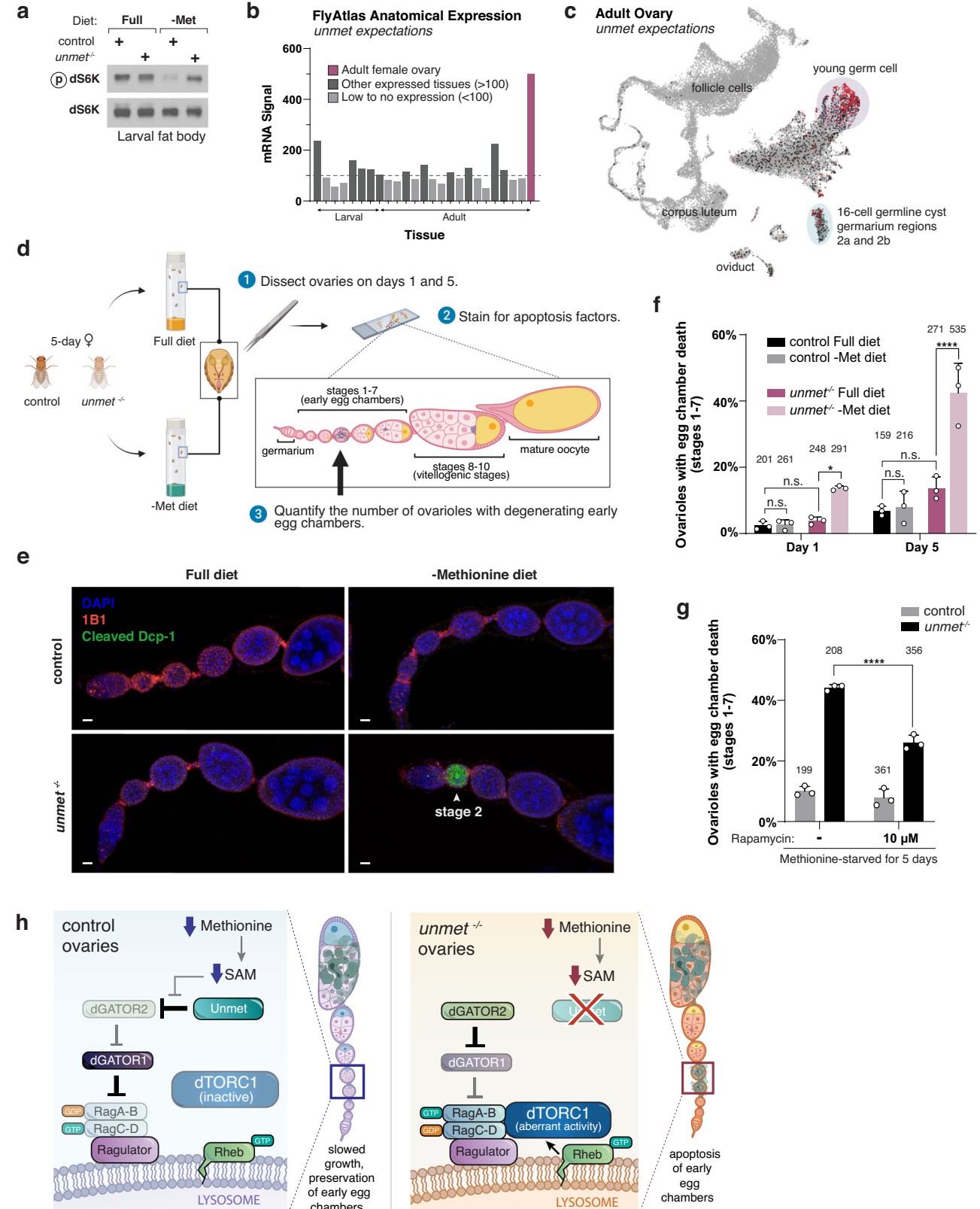

## GATOR2 guides evolution of the nutrient sensing capabilities of the mTORC1 pathway

But how does the mTORC1 pathway recruit new sensors like Unmet, especially on the relatively short time scales required for dietary adaptation? To understand how Unmet emerged as a nutrient sensor for the fly TORC1 pathway, we examined the interactions between

Unmet and GATOR2 homologs in different species. Like Unmet itself (Fig. 2c), the human homolog of Unmet, carnosine N-methyltransferase 1 (CARNMT1), co-immunoprecipitated dGATOR2 but, surprisingly, failed to bind to human GATOR2 (Fig. 5a). Similarly, the *Schizosaccharomyces pombe* homolog of Unmet interacted with dGATOR2 but not the apposite *S. pombe* SEA complex (Fig. 5b).

**Fig. 4 | Unmet maintains germline integrity in the fly ovary by suppressing dTORC1 signaling upon methionine starvation. a** Control and *unmet*[-/-] L3 larvae were transferred to either full or methionine-free holidic diets for 24 h. Dissected fat bodies were crushed and analyzed by immunoblotting for the phosphorylation state and levels of dS6K. **b** Expression of *unmet* across tissues. Anatomical expression data from the Fly Atlas, with full labels in Supplementary Fig. 4b. **c** Single-cell expression map for *unmet* in the adult ovary. HVG UMAP display of single-cell RNA-seq expression data from the Fly Cell Atlas. Cluster annotations from ref. 35. **d** Experimental setup for quantifying apoptotic early-stage egg chambers in control or *unmet*[-/-] ovaries from flies fed full or methionine-deficient diets. **e** Ovaries from female flies cultivated on the indicated diets for five days were labeled with DAPI (blue), the hu-li tai shao actin-associated antibody 1B1 (red), and cleaved *Drosophila* caspase 1 (cleaved Dcp-1 Asp216, green). The degenerating egg chamber (white arrow) is positive for cleaved Dcp-1. Scale bar, 10 μm. Full ovarioles and additional images of degenerating early egg chambers displayed in Supplementary Fig. 4d. **f** Percentage of ovarioles containing at least one dying early egg chamber for each genotype and dietary condition. Two-way ANOVA followed by Tukey's multiple comparison test; from left to right: adjusted $P = 1.0$; $P = 0.97$; *$P = 3.4 \times 10^{-2}$; $P = 0.99$; $P = 0.19$; ****$P < 0.0001$; n.s., not significant. Error bars represent the s.d. around the mean of three independent experiments. Bars are labeled with the number of ovarioles analyzed for each condition. **g** Rapamycin substantially reduces the increased apoptosis of early egg chambers in methionine-starved *unmet*[-/-] flies. Two-way ANOVA followed by Tukey's multiple comparison test; adjusted ****$P < 0.0001$. Error bars represent the s.d. around the mean of three independent experiments. Bars are labeled with the number of ovarioles analyzed for each condition. **h** Model: Unmet maintains the survival of early egg chambers during methionine starvation by detecting the absence of SAM and suppressing dTORC1 signaling. Loss of *unmet* permits inappropriately high dTORC1 activity during methionine starvation, activating a checkpoint that triggers apoptosis in early egg chambers.

Together, these data show that the fly GATOR2 complex has diverged from other GATOR2 lineages to allow for binding of Unmet and its homologs (Fig. 5c). Strikingly, they also reveal that structural changes in the dGATOR2 complex, rather than fly-specific adaptations in Unmet, directed the capture and incorporation of Unmet into the dTORC1 pathway.

Among pathways that capture new regulatory nodes by generating additional molecular interactions (Fig. 1b), this strategy, in which a conserved core component of the pathway evolves to grab an allosteric regulator, is unusual. Other signaling pathways take the opposite approach: for example, in the MAPK pathway, novel regulators establish a toehold in a pathway by targeting latent features on conserved node, followed by lengthy co-evolution[39]. To determine how the GATOR2 complex evolved a new binding surface for Unmet without compromising its existing signaling functions, we first assessed the ability of individual dGATOR2 subunits to co-immunoprecipitate Unmet (Supplementary Fig. 5a). The dWDR24, Mio, and Nup44A subcomplex was sufficient to recapitulate the interaction with Unmet; indeed, the remaining components of dGATOR2—dWDR59 and dSec13—were wholly dispensable for full binding. We therefore used the dWDR24-Mio-Nup44A subcomplex as a proxy for GATOR2 as a whole.

We then traced the evolutionary history of the Unmet-GATOR2 interaction across 11 species distributed between arthropods and vertebrates. We co-expressed homologs of Unmet and the GATOR2 tricomplex from these species in HEK-293T cells and assayed for binding (Fig. 5d, Supplementary Fig. 5b). GATOR2 acquired the ability to bind Unmet late in insect evolution, at an evolutionary branch point between honeybee (*Apis mellifera*) and mosquito (*Aedes aegypti*). The location of this branch point corresponds to the emergence of the order Diptera. To understand how the GATOR2 tricomplex recruited Unmet, we examined GATOR2 protein sequences for signatures of rapid evolution across the Dipteran branch point. Of the two unique components of the GATOR2 tricomplex, WDR24 showed no such signatures; a phylogenetic tree constructed from WDR24 sequences followed the topology of a classic species tree, in which the arthropod phylum is monophyletic, descending from a single ancestor (Supplementary Fig. 5c). By contrast, in a phylogenetic tree constructed from Mio sequences, Mio diverges so profoundly in Dipterans that homologs from other arthropods (e.g., honeybee or the crustacean *D. pulex*) cluster more closely with human and vertebrate Mio than with Dipteran proteins (Fig. 5d). Though WDR24 and Nup44A likely make additional contacts with Unmet, these data suggest that rapid evolution of Mio drove the gain-of-function in GATOR2.

To identify the molecular basis for sensor acquisition, we inspected Mio sequences for residues that are conserved in Dipterans but diverge in species that have not assimilated Unmet as a sensor. When mapped onto a recent structure of the human GATOR2 complex, these variable residues cluster on surface-exposed, flexible loops that decorate the N-terminal WD40 repeat (WDR) domain of Mio (Fig. 5e,

Supplementary Fig. 6a)[40]. While the Mio WDR domain folds into a characteristic 7-bladed beta-propeller, very few of the variable residues are involved in generating the structural fold. Instead, these residues extend from the surface of the beta-propeller and are generally not constrained by intra-complex interactions. We infer that the divergent loops define the specificity of protein-protein interactions with GATOR2. Consistent with this model, swapping the fly Mio WDR domain for a WDR domain from human Mios is sufficient to abolish binding to Unmet without disrupting formation of the dGATOR2 complex (Supplementary Figs. 6b, c). Collectively, these data argue that exposed, evolutionarily divergent loops between the structural units of the GATOR2 beta-propellers direct the fly-specific binding of Unmet.

Indeed, GATOR2 is so critical for defining regulatory inputs into the mTORC1 pathway that we can engineer artificial inputs to the human mTORC1 pathway by changing its binding behavior. Because the human GATOR2 complex cannot bind to Unmet or its human homolog CARNMT1, CARNMT1 does not regulate mTORC1 signaling in HEK-293T cells (Supplementary Fig. 7a). However, coercing a physical interaction between CARNMT1 and a core component of the mTORC1 machinery by replacing human GATOR2 with dGATOR2 allows CARNMT1 overexpression to suppress mTORC1 activity in human cells (Fig. 6a). Altering the binding capabilities of GATOR2 can thus rewire the mTORC1 pathway to respond to an enzyme that does not act as a nutrient sensor in its native cellular context. GATOR2 is therefore a flexible node that sustains regulatory complexity and innovation in the mTORC1 pathway.

Together, these findings suggest a general mechanism for the evolution of nutrient sensors without recourse to paralogous duplication (Fig. 6b). GATOR2, a conserved signaling hub for the mTORC1 pathway, can generate new binding surfaces through rapid sequence divergence of flexible loops on the beta-propellers of Mios and WDR24. Because residues on these loops do not maintain the secondary or tertiary structure of the complex, they are highly evolvable. New binding surfaces recruit pre-existing proteins, such as Unmet. If opportunistic interactions confer a selective advantage, they can be embedded into the pathway through further refinement of the interface. Strikingly, the modular structure of the GATOR2 complex, with exposed beta-propellers distributed across five different proteins, allows sequential recruitment of new pathway components without compromising existing signaling interfaces[40,41]. Given that the methyltransferase activity of Unmet is conserved from yeast to vertebrates, while its sensor role is apparently restricted to Dipterans, we infer that Unmet is an ancestral enzyme co-opted by GATOR2 for its ligand-binding capabilities. The known mammalian nutrient sensors Sestrin and CASTOR, which bind to the WDR domains of WDR24 and Mios, respectively, likely followed a similar evolutionary trajectory from enzyme to sensor (Fig. 6c). This pathway design is particularly attractive because small molecule ligand-binding is fragile and difficult to evolve de novo, in contrast to the robust evolutionary landscape for

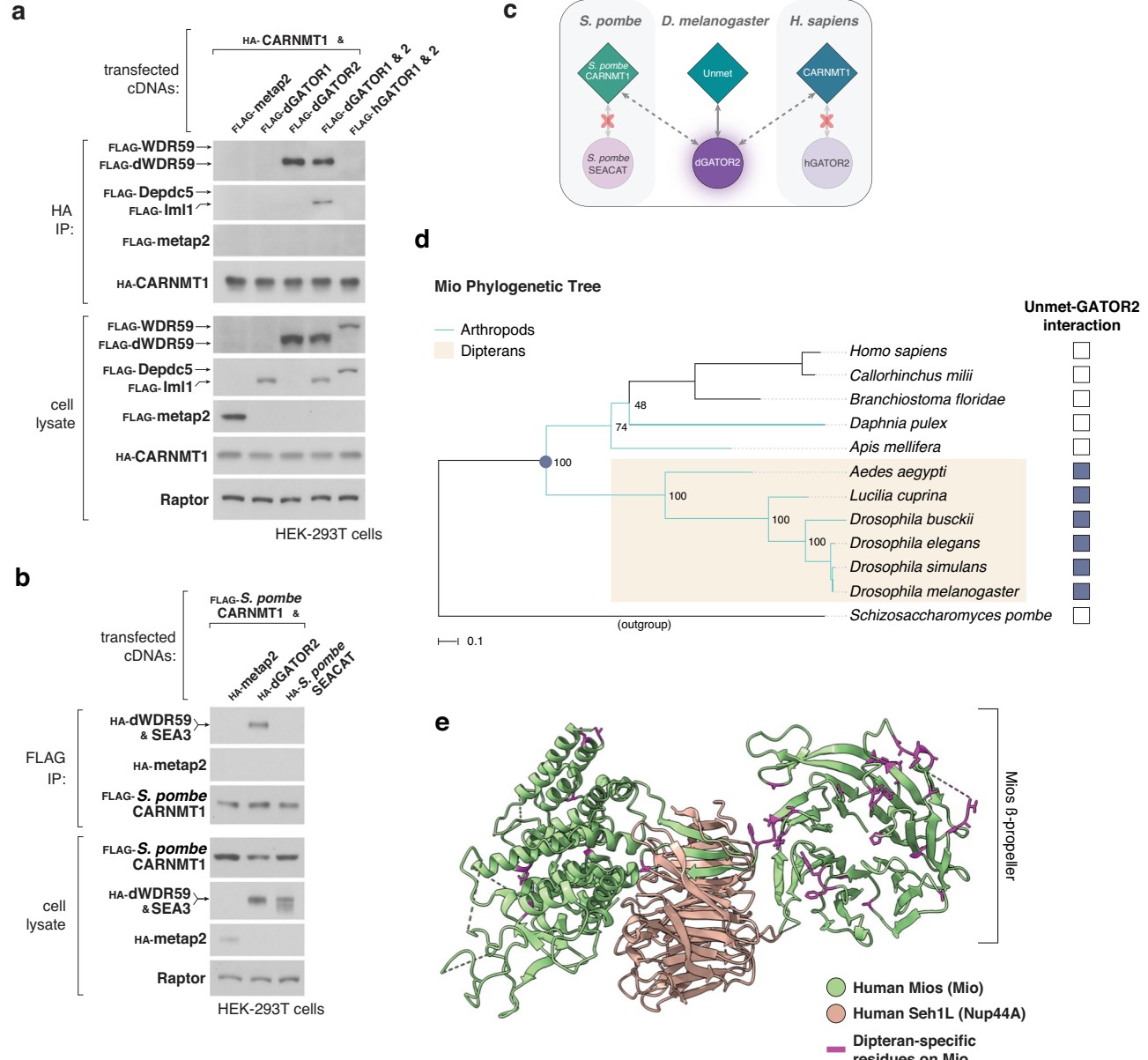

**Fig. 5 | Evolutionary adaptations in the GATOR2 complex drive the incorporation of Unmet as a nutrient sensor for the dTORC1 pathway. a** Recombinant CARNMT1, the human homolog of Unmet, interacts with dGATOR2 but not its human counterpart. Anti-HA immunoprecipitates from HEK-293T cells expressing the indicated cDNAs were analyzed as in Fig. 2b. FLAG-metap2 served as a negative control. **b** Recombinant *S. pombe* CARNMT1, the fission yeast homolog of Unmet, interacts with dGATOR2 but not the *S. pombe* GATOR2 complex (SEACAT). Anti-FLAG immunoprecipitates from HEK-293T cells expressing the indicated cDNAs were analyzed as in Fig. 2d. **c** Schematic of the interactions between homologs of Unmet and GATOR2 in three species. **d** Rapid evolution of the Mio sequence in Dipterans corresponds to the acquisition of Unmet binding. A maximum likelihood phylogenetic tree constructed using Mio protein sequences from 12 species was matched to the results of binding assays between Unmet and GATOR2 homologs, as assayed in Supplementary Fig. 5b. Mio diverged so sharply in Dipterans that arthropod sequences from outside the order cluster with vertebrate sequences, in contrast to the topology of a classical species tree, shown in Supplementary Fig. 5c. Node labels indicate bootstrap support values. Scale bar, 0.1 substitutions per site. **e** Dipteran-specific residues on Mio (magenta) are surface-exposed and map to flexible loops on the beta-propeller of Mio. Green cartoon, human Mios; orange cartoon, human Seh1L; derived from the structure of the full human GATOR2 complex (PDB: 7UHY). Dipteran-specific residues are annotated on the alignment in Supplementary Fig. 6a.

gain-of-function in protein-protein interactions[42,43]. By exploiting evolvable modules on GATOR2, the mTORC1 pathway can rapidly assimilate new sensors by repurposing proteins that already bind to a metabolite of interest while preserving information flow through the conserved core of the pathway.

## Discussion

We establish Unmet expectations as a SAM sensor for the fly TORC1 pathway. Unmet interacts with the fly GATOR2 complex in a SAM-regulated manner to control dTORC1 activity. Loss of Unmet renders the dTORC1 pathway insensitive to methionine deprivation, while expression of a mutant of Unmet that cannot bind SAM constitutively suppresses dTORC1 signaling in fly cells. Because they cannot couple SAM levels to dTORC1 activity, *unmet*−/− flies exhibit ovarian defects on methionine-free diets.

Unmet offers unique insights into the evolution of nutrient sensors in the mTORC1 pathway. Although the known mammalian nutrient sensors bear structural similarities to some bacterial proteins,

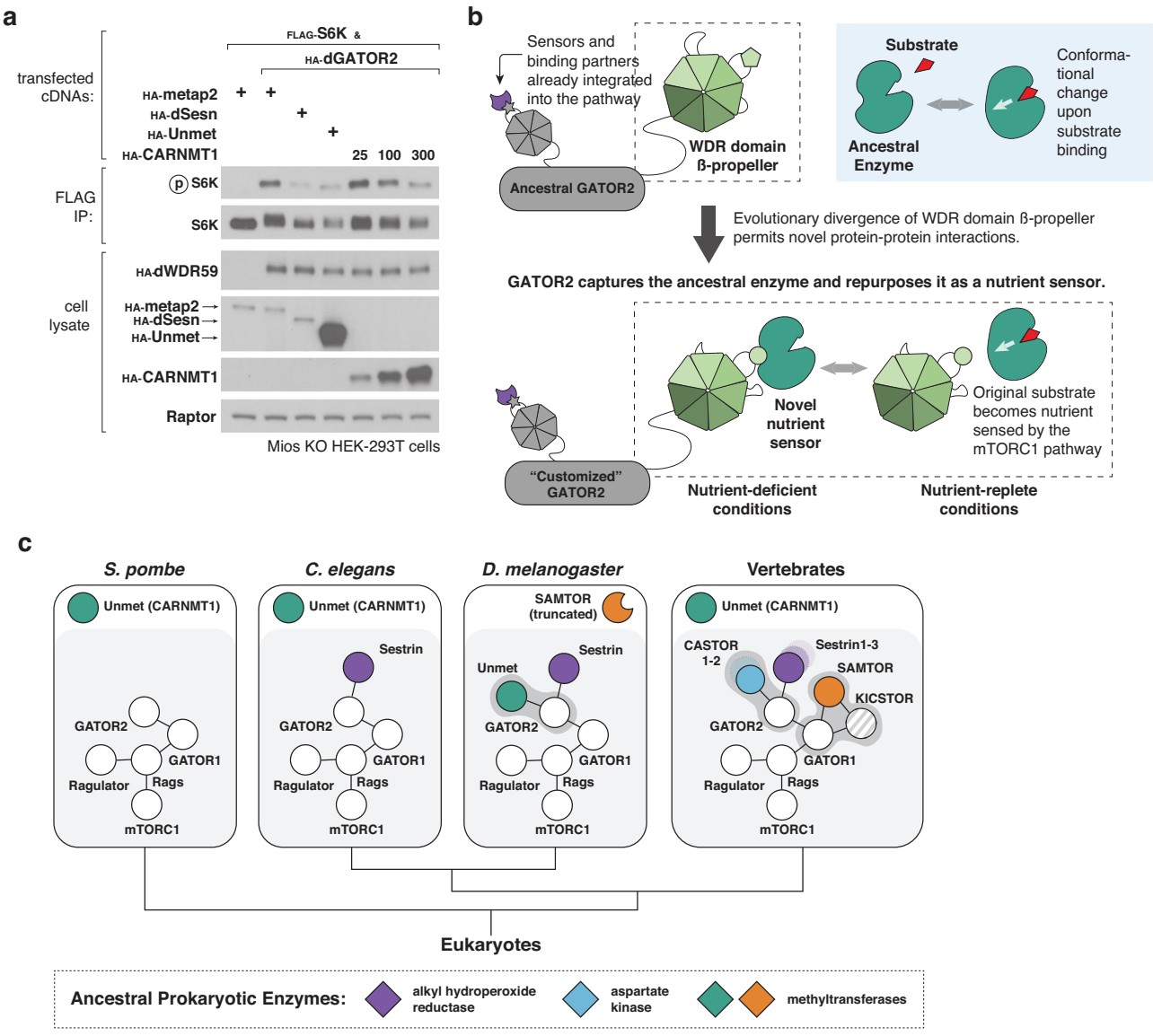

**Fig. 6 | An evolutionary mechanism to assimilate novel nutrient sensors and rewire mTORC1 signaling. a** Human CARNMT1 can act as a negative regulator of mTORC1 signaling when human GATOR2 is replaced with the fly GATOR2 complex. Mios-deficient HEK-293T cells expressing the indicated cDNAs were starved in RPMI lacking amino acids for 1 h and then restimulated with amino acids for 15 min. Anti-FLAG immunoprecipitates were analyzed as in Fig. 2d. **b** Evolutionary model for co-option of ligand-binding proteins by GATOR2. **c** Phylogenetic tree representing the evolution of nutrient sensing capabilities in the mTORC1 pathway. Conserved core components of the mTORC1 pathway are shown as white circles, connected by lines that represent protein-protein interactions. Orthologs share the same color. Dark gray blobs highlight species-restricted interactions between nutrient sensors and core components of the mTORC1 pathway. The eukaryotic nutrient sensors may ultimately share evolutionary origins with prokaryotic enzymes (shown as diamonds).

Sestrin, CASTOR1, and SAMTOR do not retain any known enzymatic activity, and their homologs have been lost in fungi and many metazoa[20,21]. As a result, they appear to emerge in higher eukaryotes as fully assimilated nutrient sensors, with few clues about their ancestral functions or evolutionary origins. Our identification of Unmet bridges that gap by showing how an independent methyltransferase, conserved across Eukarya, can be specifically repurposed in flies as a nutrient sensor. By tracing the evolutionary history of Unmet, we find that variable loops in the beta-propellers of GATOR2 can act as adapters to grab sensors from a toolkit of preexisting small-molecule-binding proteins. These data suggest an evolutionary mechanism in which ancestral enzymes are co-opted as nutrient sensors for the mTORC1 pathway (Fig. 6c).

Differences between the fly and human mechanisms of SAM sensing offer additional evidence for this model. To monitor SAM levels, flies have repurposed Unmet to bind the dGATOR2 complex,

while humans use SAMTOR and GATOR1-KICSTOR[15,18,19]. Although both of these SAM sensors have homologs in the other species—that is, human Unmet and fly SAMTOR, respectively—those homologs are not components of the mTORC1 pathway. As neither Unmet nor SAMTOR acts as a nutrient sensor in yeast or in worms, the most parsimonious explanation for these data is that SAM sensing evolved twice—once in flies and once in the vertebrate lineage—with two independent co-option events involving different methyltransferases. While we have highlighted evolvable features on GATOR2, the emergence of KICSTOR as a glue between GATOR1 and GATOR2 in vertebrates may add additional surfaces for recruitment of new mTORC1 pathway components[41]. Indeed, the evolution of SAMTOR as a sensor in vertebrates coincides with both the retention of a full-length isoform of SAMTOR and the appearance of the KICSTOR complex, suggesting that the combined GATOR1-KICSTOR binding surface is required for co-option of SAMTOR as a nutrient sensor.

Why did Dipterans and vertebrates both converge upon SAM as a metabolic regulator of the mTORC1 pathway? It is not clear what environmental triggers promoted the evolution of Unmet in the fly lineage, but one possibility is a change in diet toward less proteinaceous food sources at the evolutionary branch point between honeybees and Dipterans. The transition from diets of microorganisms or pollen, which have consistently high levels of protein, to blood or rotting fruit, where protein content is lower or variable, may have made it beneficial for Dipterans to sense SAM as a proxy for carbon or methionine[44–46]. Another possibility, raised by the mechanism of mTORC1 sensor evolution, is that SAM sensors may simply be easier to evolve than those for other nutrients. If core complexes in the mTORC1 pathway recruit sensors by developing ligand-regulated interactions with existing proteins, SAM sensors may arise more frequently because there are very many methyltransferases available for the pathway to co-opt.

Our work suggests that exaptation—repurposing existing proteins to enhance fitness in a new context—is an underappreciated theme in the evolution of sensory complexity[25]. Co-option of metabolite-binding enzymes by conserved pathway components serves as an evolutionary shortcut, exchanging the difficult task of evolving a ligand-binding site for the simpler one of evolving a new protein-protein interaction[42]. In the mTORC1 pathway, this strategy is especially effective due to the modular architecture of the very large GATOR2 complex, which insulates core signaling functions from the fitness costs of evolutionary exploration by placing hotspots for sensor acquisition in separate domains. We speculate that co-option may play a role in other conserved pathways, such as innate immune systems, that evolve receptors for new targets over short evolutionary spans.

Although Unmet offers several tantalizing hints about how living systems customize the mTORC1 pathway, full resolution of the functional organization of the pathway likely awaits the discovery of additional nutrient sensors in diverse organisms. Exploiting evolutionary insights into the mTORC1 pathway may allow us to generate artificial switches or therapeutics that regulate mTORC1 signaling with greater precision. Moreover, sensors initially characterized in other species may even be conserved in humans but may be expressed only in so-far poorly-characterized rare cell types that have specialized metabolic environments or needs.

# Methods
## Materials
Reagents were obtained from the following sources: antibody against the FLAG M2 epitope (F1804) from Millipore Sigma; antibody against Raptor (09-217) from EMD Millipore; HRP-labeled anti-mouse IgG (7076) and anti-rabbit IgG (7074) secondary antibodies from Cell Signaling Technology; antibodies against β-actin (4967), phospho-T398 dS6K (9209), Mios (13557), cleaved *Drosophila* Dcp-1 Asp216 (9578), FLAG epitope tag (14793), HA epitope tag (3724), and myc epitope tag (2278) from Cell Signaling Technology; antibody against hu-li tai shao (1B1) from the Developmental Studies Hybridoma Bank (DSHB); antibody against Depdc5 (ab185565) from Abcam; Alexa 488 and 555-conjugated secondary antibodies from Thermo Fisher Scientific. The anti-dS6K antibody was a generous gift from Mary Stewart (North Dakota State University). InstantBlue Coomassie Protein Stain was obtained from Abcam; Anti-FLAG M2 affinity gels, amino acids, SAH, carnosine, sinefungin, thiamine, riboflavin, nicotinic acid, calcium pantothenate, pyridoxine (HCl), biotin, folic acid, choline chloride,myo-inositol, inosine, uridine, methyl 4-hydroxybenzoate, potassium phosphate monobasic, sodium bicarbonate, calcium chloride hexahydrate, copper sulfate pentahydrate, iron sulfate heptahydrate, magnesium sulfate, manganese chloride tetrahydrate, zinc sulfate heptahydrate, glacial acetic acid, sucrose, and propionic acid from Millipore Sigma; DMEM, RPMI, Schneider's Medium, FreeStyle 293 Expression Medium, inactivated fetal serum (IFS), UltraPure

Salmon Sperm DNA Solution, Dynabeads M-270 Epoxy, anti-HA magnetic beads from Thermo Fisher Scientific; amino acid-free RPMI and Schneider's media lacking leucine, methionine, threonine, glutamine, phenylalanine, tryptophan from US Biologicals; [³H]-labeled SAM in sterile water (0288) from American Radiolabeled Chemicals, Inc.; SAM (13956) from Cayman Chemical; Effectene transfection reagent from Qiagen; QuickExtract DNA Extraction solution from Lucigen; EDTA-free Complete Protease Cocktail from Roche; Micropropagation Agar-Type II from Caisson Laboratories; rapamycin from LC Laboratories; Vectashield with DAPI from Vector Laboratories.

## Cell culture
HEK-293T cells obtained from ATCC (American Type Culture Collection) were cultured in Dulbecco modified Eagle's medium (Thermo Fisher Scientific) with 10% IFS (Thermo Fisher Scientific), 4.5 g/L glucose containing 2 mM GlutaMAX (Thermo Fisher Scientific), 100 IU/mL penicillin, and 100 μg/mL streptomycin. Adherent cell lines were maintained at 37 °C and 5% $CO_2$. Suspension-adapted HEK-293T cell lines were grown in FreeStyle 293 Expression Medium (Thermo Fisher Scientific) supplemented with 1% IFS, 100 IU/mL penicillin, and 100 μg/mL streptomycin. Suspension cells were grown in a Multitron Pro shaker operating at 37 °C, 8% $CO_2$, 80% humidity, and 125 rpm. *Drosophila* S2R+ cells obtained from the Perrimon lab were grown at 25 °C in Schneider's medium (Thermo Fisher Scientific) supplemented with 10% IFS (Thermo Fisher Scientific), 100 IU/mL penicillin, and 100 μg/mL streptomycin. For single-cell isolation of S2R+ cells, conditioned Schneider's media was prepared as recommended by the DRSC/TRiP (https://fgr.hms.harvard.edu/single-cell-isolation).

## Cell and tissue lysis and immunoprecipitation experiments
For lysis of S2R+ and adherent HEK-293T cells, cells were washed once with ice-cold PBS and then lysed with lysis buffer (1% Triton X-100, 40 mM HEPES pH 7.4, 10 mM β-glycerol phosphate, 10 mM sodium pyrophosphate, 2.5 mM magnesium chloride) and 1 tablet of EDTA-free protease inhibitor (Roche) per 25 mL buffer. Lysates were clarified by centrifugation at $21,000 \times g$ at 4 °C for 10 min. Dissected *Drosophila* tissues and whole flies were crushed physically utilizing a bead beater in Triton lysis buffer and processed as above.

For anti-FLAG, anti-HA, or anti-myc immunoprecipitations leading to Western blot analyses, either anti-FLAG M2 agarose beads (Millipore Sigma) or anti-HA or anti-myc-coupled magnetic beads (Thermo Fisher Scientific) were used. Beads were washed three times prior to use with Triton lysis buffer and were then incubated with the supernatant of each clarified lysate for 2 h at 4 °C. Following immunoprecipitation, beads were washed three times with Triton lysis buffer supplemented to contain 300 mM NaCl. Immunoprecipitated proteins were denatured by addition of SDS-PAGE sample buffer and boiling at 95 °C for 3 min and resolved by 8%, 10%, or 4–20% SDS-PAGE before analysis by immunoblotting. All antibodies were used at a 1:1000 dilution, except for the anti-dS6K antibody, which was used at a 1:10,000 dilution.

## Identification of Unmet by immunoprecipitation followed by mass spectrometry
S2R+ cells expressing FLAG-tagged Mio from a copper-inducible promoter at the endogenous locus were induced with 75 μM $CuSO_4$ treatment for 4 days. To generate anti-FLAG immunoprecipitates for proteomic analysis by mass spectrometry, magnetic beads bound to antibody recognizing the FLAG epitope tag were prepared in-house by coupling Dynabeads M-70 Epoxy (Thermo Fisher Scientific) to FLAG M2 antibody (Millipore Sigma), as previously described[47]. Cell lysates were prepared as described above and incubated with magnetic FLAG beads for 2 h at 4 °C. Following immunoprecipitation, beads were washed three times in lysis buffer supplemented to contain 300 mM NaCl. Proteins were eluted from the beads with the FLAG

peptide (sequence: DYKDDDDK), resolved on 4–12% NuPAGE gels (Thermo Fisher Scientific), and stained with Instant Blue (Abcam). Each gel lane was sliced into 8 pieces, followed by digestion of gel slices overnight with trypsin. The resulting digests were analyzed by mass spectrometry as described in ref. 48. This experiment was repeated three times under different conditions (in the absence of all amino acids, in the absence of leucine alone, and in the presence of all amino acids).

## Transfections

For experiments requiring transfection of DNA into HEK-293T cells, 2 million cells were plated in 10 cm culture dishes. Twenty-four hours later, cells were transfected with the appropriate pRK5-based cDNA expression plasmids using the polyethylenimine method, as previously described[49]. The total amount of DNA in each transfection was normalized to 5 μg with UltraPure Salmon Sperm DNA solution (Thermo Fisher Scientific). Forty-eight hours following transfection, cells were lysed as described above.

For experiments requiring transfection of DNA into S2R+ cells, 10 million cells were plated in 10 cm culture dishes. Cells were transfected with pGL1 or pGL2 cDNA expression plasmids using Effectene transfection reagent (Qiagen). In brief, cDNA expression plasmids added to 400 μL EC buffer were mixed with Effectene Enhancer (8 μL per 1 μg of cDNA), incubated for 5 min at RT, mixed with Effectene Reagent (10 μL per 1 μg cDNA), incubated for 10 min at RT, and then dispensed dropwise into culture dishes. Seventy-two hours after transfection and $CuSO_4$ induction (if using a pGL1 MT expression system), cells were lysed as described above.

## Amino acid starvation and restimulation of cells in culture

For experiments that required amino acid starvation, cells were washed twice with PBS and incubated in RPMI or Schneider's media lacking the designated amino acids for 90 min. To restimulate cells following starvation, an amino acid mixture prepared from individual powders of amino acids (Millipore Sigma) was added to cell culture media for 15 min.

## RNAi in *Drosophila* S2R+ cells and analysis of knockdown by qPCR

dsRNA sequences were selected from cell-screening RNAi sequences used by the DRSC. The following primer sequences, including underlined 5' and 3' T7 promoter sequences, were used to amplify DNA templates for dsRNAs targeting GFP, dSesn, Unmet, and dSAMTOR:

F-dsGFP primer:
GAATTAATACGACTCACTATAGGGAGAAGCTGACCCTGAAGTTCATCTG
R-dsGFP primer:
GAATTAATACGACTCACTATAGGGAGATATAGACGTTGTGGCTGTTGTAGTT
F-dsdSesn primer:
GAATTAATACGACTCACTATAGGGAGAGACTACGACTATGGCGAAGTGAA
R-dsdSesn primer:
GAATTAATACGACTCACTATAGGGAGATCAAGTCATATAGCGCATTATCTCG
F-dsUnmet primer:
GAATTAATACGACTCACTATAGGGAGAGCCTCCAATTTTGTCCTCAA
R-dsUnmet primer:
GAATTAATACGACTCACTATAGGGAGAGGGTTCTGTGCGTACTTGGT
F-dsdSAMTOR primer:
GAATTAATACGACTCACTATAGGGAGAAAGAAACGGTAGCGAAATGG
F-dsdSAMTOR primer:

GAATTAATACGACTCACTATAGGGAGAGATGTAGTCGATGGCCCACT

dsRNAs were produced by in vitro transcription of DNA templates using a MEGAshortscript T7 kit (Thermo Fisher Scientific).

On day one, 2 million cells S2R+ were plated into 6-well culture dishes in 1.5 mL of Schneider's media. Twenty-four hours later, cells were transfected with 2 μg of each dsRNA using an Effectene-based system (200 μL EC buffer mixed with 16 μL Effectene Enhancer and 10 μL Effectene reagent). On day four, a second round of dsRNA transfection was performed. On day five, 3 dsRNA-treated million cells were plated in 6-well culture dishes pre-coated with fibronectin. After 12 h, cells were starved for the indicated amino acids as described above.

To validate knockdown of *unmet*, *dSAMTOR*, and *dSesn*, the following primer pairs were used in qPCR reactions due to the lack of available antibodies against these proteins. *α-tubulin* was used as an internal standard. The data were analyzed by the ΔΔCt method.

F-*α-tubulin*: CAACCAGATGGTCAAGTGCG
R-*α-tubulin*: ACGTCCTTGGGCACAACATC
F-*unmet*: CTCACCTACGAGCTTGCCTG
R-*unmet*: TTGTCGCAGAGGTTGAGGAC
F-*dSAMTOR*: GACCAACGATGGGAAGGTGG
R-*dSAMTOR*: GCTCTGTAGGATTCCAGGAGT
F-*dSesn*: TCCGCTGCCTAACGATTACAG
R-*dSesn*: TTCACCAGATACGGACACTGA

## Generation of fly cells expressing endogenously FLAG-tagged proteins

To insert an N-terminal 3x-FLAG epitope tag into the *mio*, *dWDR59*, *lml1*, *unmet*, and *dSAMTOR* genes in S2R+ cells, we adapted a method described in ref. 50. Homologous recombination donor constructs were generated by PCR amplification of the following primer sequences flanking the template plasmid pRB33 (encoding a constitutively-expressed puromycin resistance marker, a copper-inducible MT promoter, and a 3x FLAG tag). Underlined sequences are complementary to the template plasmid.

*mio* HR sense:
TGCAAACTGATAACGCGACGCAATTTAGTCTGTAGTGAAAATTG
ttttttttttACATCGATGGAAAATCGGCCACGgaagttcctatactttctagaga
ataggaacttccatatg
*mio* HR antisense:
TTCCTGGCCCCAGGATACGAATTTGTCGGGAAAATGTGGAAACC
AGCTGAGTCCGTGAGTGTTGCCGCTCATaccgccgcttggagcagctgg
aga
*dWDR59* HR sense:
TTGTTTGTTGCAAAAATGGTTTAAATTCGCAGTCTTTTGCTTTTT
GAGCACTTATTAGAGTAGGACAATgaagttcctatactttctagagaatagg
aacttccatatg
*dWDR59* HR antisense:
CGGGTGCTCCTGCTCCCGGTCCACCGGCTGTTCCGCGTTCTCCC
GGACGCAGAGTCTCCGTCGGCGGCATaccgccgcttggagcagctgga
ga
*lml1* HR sense:
GCAAATGGGCAAATGTTGGAATTGAGTAAATAATTGTCCGTTGG
TTTTGCAACCACTAAGTCAACgaagttcctatactttctagagaataggaactt
ccatatg
*lml1* HR antisense:
GCAATATCCACTTTCGCTTACCGTAGGATTTGTTGCAGCCCCTC
GTATGCGTGTTCAGCTTGTACAGCTTCATaccgccgcttggagcagctg
gaga
*unmet* HR sense:
GATTACTCCCAGGATTTAAATAGCATAGATTATCGTTGAAACCG
CTGACGACGCGCCCAGgaagttcctatactttctagagaataggaacttccata
tg
*unmet* HR antisense:

GGCCAGTTGCTCGTCCATTTTAGGATGCATTGGGAACGTGGCGC
AGTCCATGCTGCTCATaccgccgcttggagcagctggaga
*dSAMTOR* HR sense:
TGTCTCATCCCTGCTGCACGCGACCCACCATTTTAGTAACACCG
AAGAAACGGTAGCGAAgaagttcctatactttctagagaataggaacttccatatg
*dSAMTOR* HR antisense:
CAGGCTTTCGTGGCAGCTCTTCACGATGCTGGCCAGGCGCTGGT
GCTCTTCAGTGGCCATaccgccgcttggagcagctggaga

U6-sgRNA fusion constructs were generated by annealing the following sequences to a U6 promoter and an optimized sgRNA scaffold as previously described[50]:

mio: cctattttcaatttaacgtcgCGATGAGCGGCAATACACAgtttaagagctatgctg

dWDR59_01: cctattttcaatttaacgtcgTAGGACAATATGCCGCCCAgtttaagagctatgctg

dWDR59_02: cctattttcaatttaacgtcgGACGCAGTGTCTCCGTGGGgttaagagctatgctg

Iml1_01: cctattttcaatttaacgtcgAGCTGAACACGCATACGCGgtttaagagctatgctg

Iml1_02: cctattttcaatttaacgtcgCAGCTTCATGTTGACTTAGgtttaagagctatgctg

Unmet_01: cctattttcaatttaacgtcgCGCGCCCAGATGAGCTCCAgtttaagagctatgctg

Unmet_02: cctattttcaatttaacgtcgGGGAACGTGGCGCAGTCCAgtttaagagctatgctg

dSAMTOR_01: cctattttcaatttaacgtcgGGTAGCGAAATGGCCACGGgtttaagagctatgctg

dSAMTOR_02: cctattttcaatttaacgtcgAACGGTAGCGAAATGGCCAgtttaagagctatgctg

S2R+ cells were transfected with dsRNAs targeting lig4 and mus308[50] to reduce non-homologous end-joining. 600,000 dsRNA-treated S2R+ cells were then seeded in 24-well culture dishes in 400 μL of Schneider's media. Twenty-four hours later, each well was transfected with the following constructs using the Effectene transfection system (100 μL EC buffer, 6 μL Effectene Enhancer, 7.5 μL Effectene reagent): 250 ng of the U6-sgRNA fusion, 250 ng pRB14 (encoding Cas9), and 250 ng of the homologous recombination donor construct.

Twenty-four hours after transfection, cells were induced with 100 μM CuSO4. On day 3 after transfection, cells were split 1:5 and replated in a 6-well dish in fresh media containing 100 μM CuSO4 and 4 μg/mL puromycin. Cells were passaged for up to 2 weeks in puromycin-containing media until control untransfected cells died. Puromycin-resistant cells were then single-cell-sorted into 96-well plates with 200 μL conditioned media. Plates were sealed with parafilm to reduce evaporation.

After 1 month of culture, individual clones were expanded. To identify clones that had an MT promoter and a 3x-FLAG tag incorporated in the endogenous gene locus, genomic DNA was extracted from each clone using QuickExtract DNA solution (Lucigen) according to manufacturer instructions. The primers indicated below were used to amplify the genomic region surrounding the insertion site:

mio_F: GTGTTTTGCGCAGCATTTTAAGTGG
mio_R: CGACTTTGCCATCCGCCAGA
dWDR59_F: TACAAACTTTTGCGACAAAATATTAGGTACAATTTTT
dWDR59_R: GTACTCTTTGCGACTGGGACATATGG
Iml1_F: GCTGACAGGGAATGCAGATTAAGTTAG
Iml1_R: GAGTACGGACGCATTTTGAAGGCA
Unmet_F: GACCCTCTTACATCCCCGTTT
Unmet_R: ACTAGCCAGATTTGGCGTGATT
dSAMTOR_F: TTATGATAAAACCAGACGGCGGC
dSAMTOR_R: GATTCCAGGAGTCGCTGCTC

Clones were validated by sequencing and by immunoblotting for the FLAG epitope after CuSO4 induction.

To restore endogenous expression of FLAG-dWDR59 and FLAG-Iml1, we transfected copper-inducible clones with 250 ng of FLP recombinase (pKF295) to flip out the puromycin resistance cassette and the MT promoter, which are flanked by FRT sites[50]. Single-cell clones with tagged protein expression under the control of the endogenous promoter were validated by sequencing and by immunoblotting for the FLAG epitope in the absence of copper.

### Generation of inducible and constitutive fly cell expression vectors

Copper-inducible pGL1 fly expression vectors for N-terminal FLAG- and HA-tagged cDNAs were generated by using EcoRI and XhoI restriction sites to insert the tag and SalI/NotI restriction sites from pRK5-FLAG or pRK5-HA vectors into a pMT-V5-His backbone (Life Technologies), followed by mutation of 2070 C > A to remove a SalI site in the backbone. Constitutive pGL2 expression vectors for N-terminal FLAG- and HA- tagged cDNAs were generated by replacing the MT promoter in pGL1 with a copia promoter using Gibson assembly.

### In vitro Unmet-dGATOR2 dissociation assay

HEK-293T cells were transiently co-transfected with the following pRK5-based cDNA expression vectors: 50 ng FLAG-dWDR59, 50 ng myc-dWDR24, 50 ng myc-Mio, 50 ng myc-Nup44A, 50 ng dSec13, and 5 ng HA-Unmet. Forty-eight hours after transfection, cells were subjected to anti-FLAG immunoprecipitations as described above. The dGATOR2-Unmet complexes immobilized on FLAG beads were washed twice in lysis buffer containing 300 mM NaCl and then incubated for 30 min in 300 μL of cytosolic buffer (0.1% Triton, 40 mM HEPES pH 7.4, 10 mM NaCl, 150 mM KCl, 2.5 mM MgCl2) with the indicated concentrations of SAM, SAH, sinefungin, or carnosine at 4 °C. The amount of Unmet that remained bound to dGATOR2 was assayed by SDS-PAGE and immunoblotting as described previously.

### Unmet protein expression and purification

To purify Rap2A and Unmet for radiolabeled SAM-binding assays, suspension-adapted HEK-293T cells grown in FreeStyle 293 Expression Medium (Thermo Fisher) supplemented with 1% IFS were transiently transfected with cDNAs encoding FLAG-tagged Rap2A or FLAG-tagged wild-type, G195D mutant, or E30A mutant Unmet on the pRK5 vector. Cells were transfected at a density of 800,000 cells/mL using 600 μg cDNA and 1.8 μg polyethylenimine per 500 mL culture. Forty-eight hours after transfection, cells were harvested, washed in ice-cold PBS, and lysed in Triton lysis buffer, as described above. Lysates were cleared by centrifugation at 40,000 × *g* for 20 min and incubated with pre-washed anti-FLAG M2 affinity gel (300 μL slurry per 500 mL culture) for 2 h at 4 °C. Beads were washed once in Triton lysis buffer, twice in Triton lysis buffer supplemented with 300 mM NaCl, and once in CHAPS buffer (0.1% CHAPS, 50 mM HEPES pH 7.4, 150 mM NaCl, 2 mM MgCl2). Proteins were eluted from the beads with 0.5 mg/mL FLAG peptide in CHAPS buffer for 2 h and concentrated with 10 kDa (for Rap2A) or 30 kDa (for Unmet) MWCO centrifugal filters (Millipore Sigma). Further purification was performed by size-exclusion chromatography on a Superose6 10/300 column (Cytiva) pre-equilibrated in CHAPS buffer supplemented with 2 mM DTT. Elution fractions were resolved by SDS-PAGE and stained with InstantBlue Coomassie Protein Stain (Abcam). Pure protein fractions were pooled and concentrated, supplemented with 10% glycerol, and snap-frozen in liquid nitrogen before storage at −80 °C.

### Radioactive SAM-binding assay

Radioactive SAM-binding assays were performed as previously reported[15]. Briefly, pre-blocked anti-FLAG M2 agarose beads (Millipore Sigma) were incubated with purified proteins (30 μL bead slurry and 10 μg protein per condition) to allow for rebinding of the proteins. The

beads were then washed and incubated for 1 h on ice in cytosolic buffer with 5 μM [³H]-labeled SAM and the indicated concentrations of unlabeled SAM, SAH, SFG, or carnosine. After this incubation, beads were aspirated dry, rapidly washed four times with binding wash buffer (cytosolic buffer supplemented with 300 mM NaCl), and resuspended in 80 μL cytosolic buffer. 15 μL aliquots from each sample were quantified using a TriCarb scintillation counter (Perkin Elmer). The SAM-binding capacity of Rap2A, wild-type Unmet, Unmet G195D, and Unmet E30A were assayed in the same experiment.

### $K_d$ calculations

The affinity of Unmet for SAM was determined by normalizing the bound [³H]-labeled SAM concentrations across three separate binding assays performed with varying amounts of unlabeled SAM. These values were plotted and fit to a hyperbolic equation (the Cheng-Prusoff equation) to estimate the $IC_{50}$ value. $K_d$ values were derived from the $IC_{50}$ value using the equation: $K_d = IC_{50}(1 + ([³H] SAM/K_d))$.

### Generation of fly cells stably expressing Unmet mutant cDNAs

For stable expression of the E30A and G195D mutants of Unmet, an N-terminal 3x-FLAG tag sequence and cDNAs encoding the indicated Unmet mutants were cloned into the pAc5-STABLE2 vector by Gibson assembly. pAc5-STABLE2 contains an mCherry cassette followed by a T2A site, followed by an eGFP cassette, a second T2A site, and a neomycin (G418) resistance cassette[51]. Tagged Unmet mutant cDNA replaced the mCherry cassette.

Three million S2R+ cells expressing copper-inducible FLAG-Unmet from the endogenous locus were plated in 6-well culture dishes and transfected with 1 μg of the stable expression vector using Effectene, as described above. Twenty-four hours after transfection, cells were transferred into Schneider's media containing 1 mg/mL G418 (Thermo Fisher Scientific) and passaged for 3–4 weeks until control untransfected cells died. Because G418 selection is often incomplete in S2R+ cells, the selected population was sorted by GFP intensity via FACS to generate a stable pool of cells expressing the mutant Unmet proteins at roughly comparable levels. To prevent silencing or changes in expression, stable pools expressing Unmet mutant cDNAs were used in dTORC1 signaling experiments within 2 weeks of isolation by FACS.

### Fly stocks, diets, and husbandry

All flies were reared at 25 °C and 60% humidity with a 12 h on/off light cycle on standard lab food (12.7 g/L deactivated yeast, 7.3 g/L soy flour, 53.5 g/L cornmeal, 0.4% agar, 4.2 g/L malt, 5.6% corn syrup, 0.3% propionic acid, 1% tegosept/ethanol).

Synthetic food was formulated and prepared as previously described[52]. For food containing 10 μM rapamycin, a 20 mM stock solution of rapamycin in ethanol was diluted 2000-fold in freshly prepared food before the agar hardened.

### Generation and validation of $unmet^{-/-}$ and $dSAMTOR^{-/-}$ fly lines

$unmet^{-/-}$ and $dSAMTOR^{-/-}$ flies were generated with CRISPR-Cas9-mediated deletion of the gene loci. Two sgRNAs with cutting sites bracketing each gene locus were cloned into the pCFD3 expression vector using the following oligonucleotide sequences[53]:

*unmet* guide 1:
sense: GTCGCCGAACCTTCGTCATCAACG
antisense: AAACCGTTGATGACGAAGGTTCGG
*unmet* guide 2:
sense: GTCGTTGGACTTGATTGTGGTGTT
antisense: AAACAACACCACAATCAAGTCCAA
*dSAMTOR* guide 1:
sense: GTCGAAGCCTGCGCCAGTTGACTA
antisense: AAACTAGTCAACTGGCGCAGGCTT

*dSAMTOR* guide 2:
sense: GTCGCTTATCTAGCTATCGTCCTG
antisense: AAACCAGGACGATAGCTAGATAAG

For each gene, both pCFD3-sgRNAs were microinjected into *y,sc,v; nos-Cas9* embryos, and emerging adults were crossed to *Lethal/FM7* (for $unmet^{-/-}$) or *Lethal/CyO* (for $dSAMTOR^{-/-}$). Progeny were screened by PCR for deletion of the whole locus using the following primers:
unmet:
F: CAGTGTAACCAGATCTAAAGTGGCGACT
R: GAGCGAGAAATTGTCCTAAAATTTGCATCC
dSAMTOR:
F: TGAATATTGGTTCTGAACGGTAAACTCGC
R: GCAATAGCATTTGTCCATTTACGACATCC

Individual *y,sc,v; unmet*$^{-/-}$ stocks were established along with *y,sc,v; +* control lines that followed the same cross scheme. Mutant stocks were sequence-verified using the primers above. To verify that *unmet*$^{-/-}$ flies no longer expressed *unmet* mRNA, total RNA was extracted from homogenized flies with TRIzol (Thermo Fisher Scientific). qPCR was performed on synthesized cDNA using a QuantStudio6 RT-PCR system (Applied Biosystems). Relative expression levels were quantified by the ΔΔCt method using the qPCR primers described above. α-tubulin served as an internal standard.

### Ovarian staining and immunofluorescence assays

To assess cell death in ovaries, 5-day-old age-synchronized, mated flies (20 females, 3 males) were flipped into vials of chemically-defined diets and maintained on those diets for 1 or 5 days. Flies were transferred to fresh vials every 2 days. Ovaries were dissected in ice-cold PBS, fixed for 20 min with 4% paraformaldehyde at room temperature, and washed three times in PBS supplemented with 0.3% Triton X-100 (0.3% PBST) for 10 min each. Samples were then blocked for 30 min (PBST, 5% BSA, 2% FBS, 0.02% NaN₃) and incubated in blocking buffer with primary antibodies overnight at 4 °C. Primary antibodies were used at the following concentrations for immunostaining: mouse anti-hts (1B1, DHSB) at 1:50, rabbit anti-cleaved Dcp-1 (Cell Signaling Technology) at 1:100. Ovaries were washed four times with PBST for 15 min and treated with Alexa 488 and 555-conjugated secondary antibodies diluted 1:400 in blocking buffer for 1 h at room temperature. After secondary antibody treatment, tissues were washed four times with PBST for 15 min before mounting in Vectashield containing DAPI (Vector Laboratories).

Ovarian images were acquired on a Zeiss LSM 710 laser-scanning confocal microscope using a ×20 objective. The Zeiss ZEN Black 2012 software package was used to control the hardware and image acquisition. Images were captured with the 405 nm, 488 nm, and 561 nm excitation lasers.

### Construction of phylogenetic trees

Homologs of mTOR (DROME01015), Unmet (DROME30051), WDR24 (DROME19416), Mios (DROME01365), and Seh1L (DROME05734) were drawn from the OMA Orthology Database, supplemented with sequences manually curated from BLASTp searches seeded by the *Drosophila melanogaster* protein sequences. Protein sequences from *Drosophila melanogaster*, *Drosophila simulans*, *Drosophila elegans*, *Drosophila busckii*, *Lucilia cuprina*, *Aedes aegypti*, *Apis mellifera*, *Daphnia pulex*, *Branchiostoma floridae*, *Callorhinchus milii*, *Homo sapiens*, and *Schizosaccharomyces pombe* were aligned using ClustalO 1.2.4. Maximum likelihood trees were constructed from protein alignments using RAxML-NG[54] with a bootstrapping cutoff of 0.03. Trees were visualized in Dendroscope 3.8.4.

### Statistical analyses

Two-tailed t-tests were used for comparison between two groups. All comparisons were two-sided, and p-values of less than 0.05 were considered to indicate statistical significance. For comparisons with two categorical factors (e.g., ovarian degeneration in flies of different

genotypes on different diets), two-way ANOVAs were used to evaluate whether the interaction term between the factors was significant, followed by post hoc analysis with Tukey-Kramer multiple comparison tests. For Tukey-Kramer multiple comparison tests, adjusted $p$-values of less than 0.05 were considered to indicate statistical significance. Sample sizes for quantitative experiments consisted of three biological replicates. No data were excluded from the analyses, and investigators were blinded to group allocation of flies during analysis and scoring of stained ovaries.

## Reporting summary

Further information on research design is available in the Nature Portfolio Reporting Summary linked to this article.

## Data availability

The proteomics data generated in this study have been deposited in the PRIDE database under accession code PXD050288. Plasmids generated in this study are available on Addgene. The data and reagents that support the findings of this study are available from the authors and the Whitehead Institute (sabadmin@wi.mit.edu) upon request. Source data are provided with this paper.

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

## Acknowledgements

We thank R.A. Weinberg and H.S. Malik for helpful discussions and critical reading of the manuscript. We gratefully acknowledge all members of the Sabatini and Perrimon Laboratories for their insights. In particular, we thank K.J. Condon and J.M. Roberts for experimental discussions and advice; M.L. Valenstein for assistance with protein purification and insights into GATOR2; B. Ewen-Campbell for advice about phylogenetic tree construction; and E. Spooner for mass spectrometric analysis of proteomics samples. Figure 4d and h were created using Biorender.com. This work was supported by grants from the NIH (R01 CA103866, R01 CA129105, and R01 AI047389 to D.M.S.; 5P01 CA120964-04 and R01 AR057352 to N.P.; T32 GM007287 and F31 CA232340 to G.Y.L.), the Lustgarten Foundation to D.M.S., and the Cystinosis Research Foundation to P.J. and N.P. N.P. is an Investigator of the Howard Hughes Medical Institute. This article is subject to HHMI's Open Access to Publications policy. HHMI lab heads have previously granted a nonexclusive CC BY 4.0 license to the public and a sublicensable license to HHMI in their research articles. Pursuant to those licenses, the author-accepted manuscript of this article can be made freely available under a CC BY 4.0 license immediately upon publication.

## Author contributions

G.Y.L. and D.M.S. conceived the study and formulated the research plan. G.Y.L. and D.M.S. interpreted experimental results with input from P.J. and N.P. G.Y.L. designed and performed biochemical experiments with assistance from R.E.B. P.J. generated the unmet$^{-/-}$ and dSAMTOR$^{-/-}$ fly strains with assistance from G.Y.L. P.J. performed larval fat body signaling experiments, while G.Y.L. performed phenotypic characterization of unmet$^{-/-}$ flies. G.Y.L. and D.M.S. wrote the manuscript. G.Y.L., P.J., N.P., and D.M.S. edited the manuscript.

## Competing interests

The authors declare no competing interests.
