## [Peer Review File · Nature Communications]

Editorial Note: This manuscript has been previously reviewed at another journal that is not operating a transparent peer review scheme. This document only contains reviewer comments and rebuttal letters or versions considered at Nature Communications.

REVIEWERS' COMMENTS

Reviewer #2 (Remarks to the Author):

This paper provides convincing data to show that the *Drosophila* Unmet protein functions as a SAM sensor. The authors also provide interesting information about the evolutionary origins of this function for Unmet. The revised manuscript has addressed many of my previous concerns. Regarding the ovarian phenotype in unmet mutants, it would strengthen the conclusions, which are based on what appears to be a loss of just a few young egg chambers, to clearly cite the work of Wei and Lilly saying that such loss can have a lasting effect on fertility. On a related note, can the authors speculate about how aberrant dTORC1 activity in unmet mutants leads to cell death? Are there clues to the mechanism of triggering apoptosis by elevated dTORC1 activity in the absence of methionine sensing?

I agree with reviewer #1 that a retraction, or at least an erratum, should be submitted for the 2017 paper from the Sabatini lab about dSAMTOR. While the data in the current manuscript correct the record, the previous Science paper should be directly acknowledged as relying on an RNAi artifact.

Reviewer #3 (Remarks to the Author):

The authors have addressed my remarks by attempting experiments and consistent rewriting. The revised manuscript is improved and convincing.

Reviewer #4 (Remarks to the Author):

The authors have provided satisfactory responses to the comments raised by this reviewer.

Reviewer #2:

This paper provides convincing data to show that the *Drosophila* Unmet protein functions as a SAM sensor. The authors also provide interesting information about the evolutionary origins of this function for Unmet. The revised manuscript has addressed many of my previous concerns. Regarding the ovarian phenotype in unmet mutants, it would strengthen the conclusions, which are based on what appears to be a loss of just a few young egg chambers, to clearly cite the work of Wei and Lilly saying that such loss can have a lasting effect on fertility.

We thank the reviewer for helpful suggestions for improving the manuscript. We have emphasized Wei and Lilly's work linking young egg chamber loss to lasting fertility defects in our revised manuscript (line 315).

On a related note, can the authors speculate about how aberrant dTORC1 activity in unmet mutants leads to cell death? Are there clues to the mechanism of triggering apoptosis by elevated dTORC1 activity in the absence of methionine sensing?

We speculate that the development of young egg chambers is subject to a checkpoint, potentially related to an appropriate ratio of germ cells to follicle cells, around stage 2-4. Evidence from the McCall group indicates that this checkpoint can fail under starvation (PMCID: [PMC2542474](https://pubmed.ncbi.nlm.nih.gov/2542474/)) and that it may be autophagy-dependent. However, it is not clear how the dTORC1 hyperactivation-dependent cell death in early egg chambers intersects with the death caused by autophagy hyperactivation. A recent paper from Youheng Wei's group suggests that both too much dTORC1 activity and too little (e.g. hyperactivation of autophagy via loss of a Rag GTPase) converge upon the same phenotypic output of apoptosis in early egg chambers through different mechanisms (<https://doi.org/10.1016/j.celrep.2023.112631>).

I agree with reviewer #1 that a retraction, or at least an erratum, should be submitted for the 2017 paper from the Sabatini lab about dSAMTOR. While the data in the current manuscript correct the record, the previous Science paper should be directly acknowledged as relying on an RNAi artifact.

We thank the reviewer for noting the strength of the data in the current manuscript. Unfortunately, many RNAi phenotypes, especially those driven by long dsRNAs, have been revealed to be artifacts by subsequent papers using CRISPR knockouts, but they do not usually result in errata as the papers accurately report results obtained with what were considered state of the art reagents at the time (several examples are reviewed in PMCID: [PMC5206767](https://pubmed.ncbi.nlm.nih.gov/25206767/)). Under current norms, errata are made to a paper when a substantial mistake was made by the authors in analysis or presentation of the data, which is not the case here. We want to again emphasize that the RNAi artifact affects the interpretation of a single panel but certainly not the overall conclusions of the 2017 SAMTOR paper, which is concerned with the discovery and characterization of human SAMTOR, not (with the exception of Figure 4d) dSAMTOR. We have consulted the first and corresponding authors on the 2017 paper, and they do not feel that an erratum is necessary in this case. However, we are open to discussing this further, particularly if

there is a change in policy that mandates corrections to all papers containing artifacts from the off-target effects of RNAi reagents.